New evidence for the co-occurrence of two genera of Paleoparadoxiidae (Mammalia, Desmostylia) from the Middle Miocene of Japan: insights into taxonomic status and paleodiversity in Desmostylia

Asai Yuma yuma.asai.54@gmail.com 1 2
Ando Tatsuro 3
Sawamura Hiroshi 3
Hayashi Shoji 1 4
1 Faculty of Biosphere-Geosphere Science, Okayama University of Science , Okayama , Japan
2 Graduate School of Science and Technology, University of Tsukuba , Tsukuba , Ibaraki , Japan
3 Ashoro Museum of Paleontology , Ashoro , Hokkaido , Japan
4 Division of Materials and Manufacturing Science, Graduate School of Engineering, Osaka University , Suita , Osaka , Japan
Ward Eric
Electronic publication date: 2025 Jul 31
Publication date: 2025
Volume: 13
Electronic Location ID: e19578
Received 2025 Apr 3; Accepted 2025 May 19
Copyright: ©2025 Asai et al.
Copyright year: 2025
Copyright holder: Asai et al.
License: This is an open access article distributed under the terms of the Creative Commons Attribution License, which permits unrestricted use, distribution, reproduction and adaptation in any medium and for any purpose provided that it is properly attributed. For attribution, the original author(s), title, publication source (PeerJ) and either DOI or URL of the article must be cited.
License URL: https://creativecommons.org/licenses/by/4.0/

Keywords: Marine mammal, Desmostylia, Paleoparadoxiidae, Desmostylidae, Diversity, Miocene

Funding: The Japan Society for the Promotion of Science KAKENHI 23K05910 This work was supported by the Japan Society for the Promotion of Science KAKENHI (grant Number 23K05910). The funders had no role in study design, data collection and analysis, decision to publish, or preparation of the manuscript.

==============================
Desmostylia, an extinct order of marine mammals, includes two major families: Paleoparadoxiidae and Desmostylidae. Within Paleoparadoxiidae, three genera—Archaeoparadoxia, Paleoparadoxia, and Neoparadoxia—have been identified, with Paleoparadoxia being the only genus found on both coasts of the North Pacific Rim. In Akan, Hokkaido, Japan, one of the largest Paleoparadoxia fossil assemblages in the world has been discovered from the Middle Miocene Tonokita Formation. Previous studies suggested the presence of two congeneric species of Paleoparadoxia, while recent taxonomical revisions raise the possibility that two genera, Paleoparadoxia and Neoparadoxia, were present in Akan. Here, we describe three paleoparadoxiids specimens from Akan, which consist of two partial crania and a mandible. Based on morphological comparisons and phylogenetic analysis, we identified these specimens as Paleoparadoxia sp. and Neoparadoxia sp. This represents the first record of two genera of Paleoparadoxiidae in the same locality and horizon, with the potential to provide valuable insights into cladogenesis and morphological diversification within this family. In addition, the Akan specimens exhibit mosaic characteristics of Paleoparadoxia and Neoparadoxia, suggesting that a reassessment of their morphological features for taxonomic identification and phylogenetic analysis is required. To better understand species-level diversity patterns in Desmostylia, we conducted stage-binned analysis and richness curve analysis. Our analysis revealed three significant points in their diversification history: (1) Desmostylidae reached peak diversity at the Oligocene-Miocene boundary, coinciding with a glacial event; (2) Paleoparadoxiidae achieved peak diversity during the Middle Miocene global warming event; (3) both families declined in diversity and went extinct during the Middle to Late Miocene global cooling event. These findings indicate that desmostylian diversity could have been closely linked to climatic events, with the differing peak diversities of Paleoparadoxiidae and Desmostylidae reflecting their respective preferences for warmer and cooler climates. Our analysis provides a valuable baseline for understanding the diversity and evolutionary history of Desmostylia.

Introduction

Desmostylia is an extinct order of quadrupedal marine mammals from Oligocene and the Miocene deposits of the northern Pacific Rim (e.g., Domning, Ray & McKenna, 1986; Inuzuka, Domning & Ray, 1994). This group is divided into two major clades, Desmostylidae and Paleoparadoxiidae (Osborn, 1905; Reinhart, 1959; Matsui & Tsuihiji, 2019). Paleoparadoxiidae forms a monophyletic group with three genera and four species: Archaeoparadoxia weltoni, Paleoparadoxia tabatai, Neoparadoxia repenningi and Neoparadoxia cecilialina. These species lived from the Late Oligocene to the Late Miocene and were primarily discovered in the United States (Barnes, 2013). Among them, only Paleoparadoxia is known from both the Japan and North American coasts, with over 30 fossil localities reported in Japan (Inuzuka, 2005; Matsui & Kawabe, 2015).

In Akan, Hokkaido, Japan, one of the largest Paleoparadoxia fossil assemblage in the world has been found from the Middle Miocene Tonokita Formation (Kimura et al., 1998). Previous studies suggested the presence of both a large and a small Paleoparadoxia species in Akan (Akan vertebrate fossil assemblage research group, 2000; Akan vertebrate fossil assemblage research group, 2002; Inuzuka, 2005; Fig. 1A). The Akan is the only site where two Paleoparadoxia are suggested to co-occurrence in the same stratigraphic horizon, potentially providing a key fossil record for understanding cladogenesis within this group. However, recent studies reidentified the large Paleoparadoxia species as Neoparadoxia, a newly established genus (Barnes, 2013). While this study did not mention the Akan specimens, the most recent taxonomic hypotheses raise the possibility that both Paleoparadoxia and Neoparadoxia may have been present in Akan (Matsui & Tsuihiji, 2019; Fig. 1B).

Figure 1 Previously-proposed hypothesis of the phylogeny for the Desmostylia.

Topologies have been modified from previous studies. The species of Paleoparadoxiidae suggested to have existed in Akan are colored in red. The numbers in parentheses indicate type specimen numbers. (A) The topology of Inuzuka (2005), (B) the topology of Matsui & Tsuihiji (2019), and highlighting Paleoparadoxiidae and Desmostylidae.

Repeated taxonomic revisions have hindered comprehensive analyses of desmostylians diversity. Some studies suggest that the body size and morphological diversity of Paleoparadoxiidae had increased from the Middle to Late Miocene (Inuzuka, 2005; Pyenson & Vermeij, 2016; Matsui, Valenzuela-Toro & Pyenson, 2022). However, it remains debated whether these changes represent cladogenesis or intraspecific variation. Additionally, research on the diversification and evolutionary drivers of Desmostylia is limited, with Berta & Lanzetti (2020) providing the only comprehensive study to date. The Akan specimens offer significant potential to enhance our understanding of their evolutionary history because they were discovered during the Middle Miocene, when evolutionary changes in Paleoparadoxiidae began.

Here, we describe three paleoparadoxiids specimens from Akan (AMP AK960241, AMP AK970253 and AMP AK000247). Our study represents the first record of the co-occurrence of Neoparadoxia and Paleoparadoxia, shedding new light on the taxonomic status of Paleoparadoxiidae. In addition, we conducted two types of paleodiversity analyses to clarify the diversification patterns in Desmostylia. Comparing paleodiversity patterns is widely recognized as a valuable approach for examining evolutionary drivers (e.g., Ando & Fordyce, 2014; Boessenecker & Churchill, 2018; Berta & Lanzetti, 2020). In this study, we apply this approach to analyze the diversification dynamics of Desmostylia, assess their potential drivers, and discuss possible biases that should be taken into consideration in future studies.

Material and Methods

Geological setting

The Akan specimens were found on the bank of Chichappupon river in Akan, Kushiro City, Hokkaido, Japan (Fig. 2; 43°11′42″N, 144°05′42″E). The Middle Miocene Tonokita Formation crop out in this region. The Tonokita Formation comprise conglomerate to coarse sandstone containing diatoms, pollen fossil, molluscan shells and marine vertebrates. The marine mammal fossil assemblage was found in pebbly sandstone in the Tonokita Formation, which occur in channel deposits sometimes incised into the under layer. These fossils are mostly fragmented, and the molluscan fossils are disarticulated, suggesting secondarily deposition. The marine mammals include Desmostylia (Desmostylus sp. and Paleoparadoxia sp.), Pinnipedia and Odontoceti (Okazaki et al., 1972; Kimura et al., 1998; Akan vertebrate fossil assemblage research group, 2000; Akan vertebrate fossil assemblage research group, 2002). The depositional environment of the Tonokita Formation is interpreted as upper shelf or lower-shoreface (Urabe & Hoyanagi, 2004). The paleoclimate was estimated based on marine mollusks and pollen fossil. The molluscan fauna (Atsunai Fauna; e.g., Glycymeris idensis, Swiftopecten swiftii, Chlamys cosibensis, Masudapecten sp., Felaniella usta, Serripes groenlandicus, Mercenaria yiizukai, Spisula onnechiuria, Crepidula jimboana, Nucella tokudai) suggests a cool temperate climate (Suzuki, Kimura & Tanaka, 1999; Suzuki, 2000). Similarly, pollen fossils indicate a temperate climate, characterized by a mixed forest of Taxodiaceae, Alnus and Betula (Igarashi, Yahata & Kimura, 2000; Yahata & Kimura, 2000). Diatom fossils from the Tonokita Formation indicate an age range from 15.9 Ma to 14.9 Ma, corresponding to the Denticulopsos lauta zone (Urabe, Akiha & Hoyanagi, 2003).

Figure 2 Geographic and geological context of paleoparadoxiids specimens from Akan.

(A) The locality of Neoparadoxia sp. (AMP AK960241) and Paleoparadoxia sp. (AMP AK970253, AMP AK000246), modified from Kimura et al. (1998) and the topographic maps from the Geospatial Information Authority of Japan (GSI). (B) Diatom zone and stratigraphic diagram, modified from Urabe, Akiha & Hoyanagi (2003) and Urabe & Hoyanagi (2004). (C) Stratigraphic column of the Tonokita Formation exposed Chichappupon River, Akan, Hokkaido, Japan, modified from Kimura et al. (1998) and Igarashi, Yahata & Kimura (2000).

Materials

The Akan specimens were discovered during five excavation surveys conducted between 1996 and 2000. Although the remains are isolated, they include various parts of the body, such as cranial and limb bones, from individuals ranging from juveniles to adults (Akan vertebrate fossil assemblage research group, 2000; Akan vertebrate fossil assemblage research group, 2002). In this study, we focused on three paleoparadoxiids specimens from the Akan, consists of two partial crania and a mandible (AMP AK960241, AMP AK970253, AMP AK000247). AMP AK960241 and AMP AK970253 were excavated in 1996 and 1997, respectively. These specimens were previously identified as Paleoparadoxia sp. In contrast, AMP AK000247, discovered in 2000, was assigned to P. tabatai (Akan vertebrate fossil assemblage research group, 2000; Akan vertebrate fossil assemblage research group, 2002). All three specimens were collected from the same site and stratigraphic horizon. These specimens are currently housed at the Ashoro Museum of Paleontology (AMP).

Phylogenetic analysis

The phylogenetic position of the Akan specimens described here was analyzed based on the data matrix of Matsui & Tsuihiji (2019), adding the three Akan specimens and four new characters. These newly added characters are as follows: character 109, parietal inclined to cranial (0), slightly inclined to cranial (1), not inclined (2); character 110, paroccipital process of ventral side: inclined medially (0), not inclined or expand laterally (1), not elongate (2); character 111, a few small foramina and sulci in supraorbital process: present (0), only foramina present (1), absent (2); character 112, mandibular fossa or mandibular condyle): long axis is oriented medio-laterally (0), oriented anterolaterally (1). All 112 characters were given equal weight in the analysis. The dataset included 19 operational taxonomic units (OTUs). Outgroups were selected to reflect the three major phylogenetic hypotheses regarding the affinities of Desmostylia, following Matsui & Tsuihiji (2019): the Perissodactylamorpha hypothesis (Cooper et al., 2014; Rose et al., 2014; Rose et al., 2020), the Afrotheria hypothesis (Domning, Ray & McKenna, 1986) and the Paenugulatomorpha hypothesis (Gheerbrant, Filippo & Schmitt, 2016). The outgroups used were Anthracobune spp. (a primitive perissodactyl, coding based on Cooper et al. (2014)), Pezosiren portelli (a primitive sirenian, with coding based on Andrews (1904) and Andrews (1906)), and Moeritherium spp. (a primitive proboscidean, with coding based on Holroyd et al. (1996), and Delmer et al. (2006)). The ingroup taxa include all 13 known species of Desmostylia, as well as the three desmostylids specimens from Akan (AMP AK960241; AMP AK970253; AMP AK000246).

Phylogenetic analysis was performed using PAUP Version 4.0a10 for Windows (Swofford, 2002) with a heuristic search option, optimized by the accelerated transformation criterion (ACCTRAN) using the tree-bisection-reconnection (TBR) branch swapping and a simple addition sequence. Bootstrap analysis was conducted after a full heuristic search with 10,000 replications. To incorporate time constrains into the phylogenetic tree, we applied the geoscalePhylo function in the paleotree package (Bapst, 2012) for the R version 4.4.1 (R Core Team, 2024).

Estimation of Desmostylian diversity

In this study, we conducted two analyses—Stage-binned analysis and Richness curve analysis—to elucidate the diversity changes within two major desmostylian families, Paleoparadoxiidae and Desmostylidae. These analyses are based on the desmostylian fossil occurrence (Supplemental Information 1) and our phylogenetic analysis, which suggests that Paleoparadoxia sp. and Neoparadoxia sp. in Akan represent distinct species from those currently recognized.

For the stage-binned analysis, we followed the methods of Boessenecker & Churchill (2018) and Berta & Lanzetti (2020). This analysis examined over 200 desmostylian fossil localities and their stratigraphic ages, determined through a comprehensive literature review of chronostratigraphic data (Supplemental Information 1). The number of desmostylian species within each chronostratigraphic stage was counted under the assumption that species persisted continuously throughout their confirmed temporal ranges. Diversity plots were obtained by counting the valid species for each time bin.

For the time-constrained richness curve analysis, we employed methods based on Magallanes et al. (2018) to illustrate the minimum and maximum counts of known and inferred lineages (richness) of Desmostylidae and Paleoparadoxiidae throughout their evolutionary history. This approach, accounted for chronostratigraphic uncertainty and ghost lineages. For each taxon, we estimated a minimum and maximum possible age of occurrence, building upon estimates presented in recent studies (Supplemental Information 1). Using these age ranges, we constructed a time-constrained 50% majority-rule consensus tree to calculate the minimum and maximum lineages counts at 0.1 Ma interval (35.0 Ma to 5.0 Ma, 300 time slices). The maximum lineage count was calculated based on the assumption that each taxon existed from the earliest to the latest possible age of the rock units. For three taxa known from multiple formations (P. tabatai, Cornwallius sookensis, Desmostylus hesperus), the combined maximum age ranges of these formations were used to determine the broadest possible temporal span. The minimum lineage count was estimated by assuming originations and extinctions within the possible range that resulted in the minimized the number of lineages at any given time. In most cases, this involved counting ghost lineages, as a taxon within its estimated range could be presumed either extinct or not yet originated at any given time. Maximum and minimum age estimates for each 0.1 Ma bin among the trees was included in the curves.

Systematic paleontology

DESMOSTYLIA Reinhart, 1953	
DESMOSTYLOIDEA Osborn, 1905 (sensuMatsui & Tsuihiji, 2019)	
PALEOPARADIXIIDAE Reinhart, 1959 (sensuMatsui & Tsuihiji, 2019)	
NEOPARADOXIA Barnes, 2013	
Neoparadoxia sp.	
(Figs. 3–6, Table 1)	

Emended diagnosis of genus: Neoparadoxia can be distinguished from other desmostylians by its paroccipital process, which is massive, elongated, thickened, and extends laterally. It differs from other paleoparadoxiids in possessing following characters: extremely large body size; a proportionally large and deeply concave temporal fossa; a large mandibular fossa.

Materials: AMP AK960241 (cranium, Figs. 3–6, Table 1).

Locality: The riverbank where the Chichappupon river in the Akan, Kushiro city, Hokkaido, Japan (Fig. 2; 43°11′42″N, 144°05′42″E).

Age and horizon: Tonokita Formation, Middle Miocene, Langhian (15.9-14.9 Ma; Figs. 2B, 2C).

Description

AMP AK960241 (posterior part of the cranium): AMP AK960241 preserves supraoccipital, exoccipital, basioccipital, basisphenoid, parietal and squamosal (Figs. 3–6). Although the parietal and zygomatic arch are heavily damaged, the exoccipital is perfectly preserved. This specimen, consisting of three fragments, is extremely large—significantly larger than P. tabatai and comparable in size to N. cecilialina (Table 1). This suggests that it is among of the largest paleoparadoxiid cranium ever discovered from Japan.

The occipital region is nearly complete on the right side (Figs. 3, 5 and 6). The external occipital crest is distinct and the connection to the nuchal crest differs from the condition in Behemotops katsuiei (Inuzuka, 2000), and the external occipital protuberance is slightly developed. The occipital condyle is preserved only at its base. The paroccipital process is massive, medially inclined, expand laterally and the tip is thickened. The ventral condylar fossa is well-developed. The hypoglossal foramen is partially damaged and can be observed at the boundary between the exoccipital and the basioccipital, at the base of the paroccipital process. The stylomastoid foramen opens ventrally on the lateral surface of the exoccipital, posterior to the mandibular fossa. The pharyngeal and the muscular tubercles on the basioccipital are prominent.

The basisphenoid is nearly completed (Fig. 4). The lacerate foramen is widely open between the squamosal and basisphenoid, and the oval foramen is also situated between these two bones.

The parietal, preserved in its posterior portion (Figs. 3, 5 and 6), is slightly inclined cranially. The nuchal crest is pronounced and nearly V-shaped, while sagittal crest strongly developed—unlike Desmostylus and Ashoroa laticosta (Inuzuka, 1988; Inuzuka, 2000). The parasagittal crests are extremely narrow and positioned close to each other. The temporal fossa is notably large and the basisphenoid-occipital suture is completely closed.

The squamosal is partially preserved (Figs. 3–6). The squamosal foramina are present the base of zygomatic arch and near the nuchal crest, with the posterior one being the largest. The number of foramina (three to four) is consistent with P. tabatai and N. cecilialina. The foramina positions are asymmetrical as in other desmostylians. A passage connecting the anterior external auditory meatus to the skull roof is present, unlike in Desmostylidae (Clark, 1991). The mandibular fossa is missing anteriorly, but the preserved portion suggests that it was extremely large. The long axis of the mandibular fossa is oriented anterolaterally, a feature shared with other paleoparadoxiids, Behemotops and Seuku emlongi (Domning, Ray & McKenna, 1986; Inuzuka, 2000; Inuzuka, 2005; Barnes, 2013; Beatty & Cockburn, 2015).

The tympanic region is exposed due to damage to the zygomatic arch and lateral squamosal (Figs. 3 and 5). The external auditory meatus opens posteriorly on lateral surface of the zygomatic arch and is situated posterodorsally to the mandibular fossa. The epitympanic sinus lies slightly dorsally adjacent to the external auditory meatus. This positional relationship is similar to the condition in D. hesperus and C. sookensis, suggesting that it may be a shared morphological character within Desmostylia (Clark, 1991; Uno & Kimura, 2004; Beatty, 2009). The tympanic bone, which connects to external auditory meatus, forms a rounded equilateral triangle and measuring approximately 2.5 cm per side. The cochlea is exposed medially and features a three-tiered spiral structure measuring approximately 14.4 mm dorsoventrally, filled with sediments. The postzygomatic foramen, located adjacent to the epitymapanic sinus, is wide and shallow, consistent with other desmostylians (Ijiri & Kamei, 1961; Inuzuka, 1988; Beatty, 2009).

PALEOPARADIXIIDAE Reinhart, 1959 (sensuMatsui & Tsuihiji, 2019)	
PALEOPARADOXIA Reinhart, 1959	
Paleoparadoxia sp.	
(Figs. 7–12, Tables 2 and 3)	

Emended diagnosis of genus: Paleoparadoxia differs from all other desmostylians in possessing the following characters: the zygomatic arch of squamosal inclined to caudally and not broadened dorsoventrally; the coronoid crest of the dentary curved anteriorly. It differs from Neoparadoxia in possessing following characters: the supraorbital process is slightly expanded laterally; the dorsal surface of cranium between supraorbital processes is not depressed; the orbit is positioned slightly dorsally; the mandibular symphysis is slightly rotated anteroventrally.

Figure 3 Dorsal view of the skull of Neoparadoxia sp. (AMP AK960241).

(A) Photo. (B) Corresponding line drawing with anatomical interpretations. Hatched areas indicate broken surfaces.

Table 1 Measurements (in mm) of skull of Neoparadoxia sp. (AMP AK960241).

	AMP AK960241	P. tabatai	N. cecilialina	
Total length (sagittal direction)	138a	482	553	
Cranium height	187	139	165	
Paroccipital width	353b	232	328b	
Zygomatic width	316a,b	239	324	
Width across occipital condyle	146a,b	124	162	
Notes.

Modified from Inuzuka (2005) and Barnes (2013).

a Broken, measured as preserved.

b Broken, the measurements are doubled for presentation.

Figure 4 Ventral view of the skull of Neoparadoxia sp. (AMP AK960241).

(A) Photo. (B) Corresponding line drawing with anatomical interpretations. Hatched areas indicate broken surfaces.

Figure 5 Right lateral view of the skull of Neoparadoxia sp. (AMP AK960241).

(A) Photo. (B) Corresponding line drawing with anatomical interpretations. Hatched areas indicate broken surfaces.

Materials: AMP AK970253 (cranium, Figs. 7–9, Table 2), and AMP AK000247 (mandible, Figs. 10–12, Table 3).

Figure 6 Caudal view of the skull of Neoparadoxia sp. (AMP AK960241).

(A) Photo. (B) Corresponding line drawing with anatomical interpretations. Hatched areas indicate broken surfaces.

Figure 7 Dordal view of the skull of Paleoparadoxia sp. (AMP AK970253).

(A) Photo. (B) Corresponding line drawing with anatomical interpretations. Hatched areas indicate broken surfaces.

Figure 8 Ventral view of the skull of Paleoparadoxia sp. (AMP AK970253).

(A) Photo. (B) Corresponding line drawing with anatomical interpretations. Hatched areas indicate broken surfaces.

Figure 9 Lateral view of the skull of Paleoparadoxia sp. (AMP AK970253).

(A) Photo. (B) Corresponding line drawing with anatomical interpretations. Hatched areas indicate broken surfaces.

Figure 10 Dorsal view of the mandible of Paleoparadoxia sp. (AMP AK000247).

(A) Photo. (B) Corresponding line drawing with anatomical interpretations. Hatched areas indicate broken surfaces.

Locality: The riverbank where the Chichappupon river in the Akan, Kushiro city, Hokkaido, Japan (Fig. 2; 43°11′42″N, 144°05′42″E).

Age and horizon: Tonokita Formation, Middle Miocene, Langhian (15.9–14.9 Ma; Figs. 2B, 2C).

Description

AMP AK970253 (posterior part of the cranium): AMP AK970253 preserves the frontal, basisphenoid, parietal, squamosal, and vomer (Figs. 7–9). Its size is comparable to that of the corresponding part in P. tabatai (Table 2). The lambdoid suture is detached while the squamosal suture is fused, as seen in adult P. tabatai, thus we identified this specimen as a relatively mature one.

The frontal is heavily damaged in its middle section (Figs. 7 and 9). The orbits projected slightly from the frontal, and the orbital margins are thickened, porous, and rounded. The supraorbital process is weakly developed laterally and dorsally, similar to P. tabatai, but less prominent than in A. weltoni and N. cecilialina (Clark, 1991; Barnes, 2013). Nutrient foramina and radiating sulci are supraorbital process. The space between the supraorbital processes is heavily damaged but remains flat.

The basisphenoid is preserved (Fig. 8). The left tympanic bulla is partially-intact anteriorly but missing posterior and medial portion. The shape of the tympanic bulla is elliptical, with its long axis is oriented posterior-laterally. The oval foramen is visible posteriorly on both sides of the sphenoid bone. The pterygoid process is preserved only at its base.

The vomer is heavily damaged, but its suture with the basisphenoid is fully fused.

The parietal (Figs. 7 and 9) is slightly inclined cranially, with a weakly developed sagittal crest and widely spaced parasagittal crests. The nuchal crest is pronounced and nearly V-shape—unlike Desmostylus and A. laticosta (Inuzuka, 1988; Inuzuka, 2000).

The squamosal is preserved (Figs. 7–9). The squamosal foramina are present at the base of zygomatic arch and near the nuchal crest, with the posterior foramen being the largest. The number of foramina (three to four) is consistent with P. tabatai and N. cecilialina. The positions of these foramen are asymmetrical, a common character among desmostylians (Ijiri & Kamei, 1961; Barnes, 2013; Inuzuka, 1988). A passage connecting the anterior external auditory meatus to the skull roof is observed, unlike in Desmostylidae (Clark, 1991). The mandibular fossa is flat and faced outward. The long axis of the mandibular fossa is oriented anterolaterally, a character shared with other paleoparadoxiids, Behemotops and S. emlongi (Domning, Ray & McKenna, 1986; Inuzuka, 2000; Inuzuka, 2005; Barnes, 2013; Beatty & Cockburn, 2015). The zygomatic arch of squamosal is caudally inclined and relatively thin. The posterior end of the jugal is retracted and does not reach the mandibular fossa as seen in P. tabatai but not in A. weltoni (Clark, 1991). The postzygomatic foramen is wide and shallow, consistent with other desmostylians (Ijiri & Kamei, 1961; Inuzuka, 1988; Beatty, 2009). A funnel-shaped external auditory meatus opens posteriorly behind the zygomatic arch and the mandibular fossa.

AMP AK000247 (mandible): AMP AK000247 is mostly preserved, with the exception of anterior portion of the mandible (Figs. 10–12). It is slightly smaller in size compared to the P. tabatai and A. weltoni (Table 3). The mandibular body is straight in medial and lateral view, with the upper margin curving slightly posteriorly —unlike the sigmoid shape observed in Desmostylus (Inuzuka, 2000; Inuzuka, 2005). The mental foramina are present on the lateral side of the mandibular body while the interalveolar margin of the mandible is nearly straight, featuring a large depression on the lingual side. The mandibular symphysis is rotated anteroventrally, becoming approximately horizontal and aligning incisors and canines anteriorly, as seen in Paleoparadoxiidae (Barnes, 2013). The masseteric fossa is wide and shallow, while the mylohyopid groove is distinct. The mandibular ramus and the mandibular body form a nearly right angle. The mandibular foramen is oval-shaped, measuring 11.9 mm in short diameter and 23.1 mm in long diameter. The mandibular notch is shallow and the mandibular angle is rounded, bending nearly perpendicularly. The long axis of the mandibular condyle is oriented anterolaterally, as observed in Paleoparadoxiidae, Behemotops and S. emlongi (Domning, Ray & McKenna, 1986; Inuzuka, 2000; Inuzuka, 2005; Barnes, 2013; Beatty & Cockburn, 2015). The coronoid crest (anterior margin of coronoid process) curves anteriorly, with a small angle between the anterior and posterior margin of the coronoid process.

The teeth are not preserved but the alveoli for p1, p2, p4, m1, and m2 are preserved. The alveolus of p1 is elongated sagittal direction, with a diastema present between p1 and p2. The p2 and p4 teeth are closely positioned near the distal end of the mandibular symphysis m1 and m2 are slightly compressed laterally, each displaying two distinct roots. m2 is larger in the sagittal direction compared to m1. The absence of the m3 alveolus suggests that this specimen is a large juvenile.

Result

Phylogenetic analysis

Our phylogenetic analysis resulted in 136 most parsimonious trees with a tree length 206 after 311,144 rearrangements were attempted. The consistency index (CI) is 0.623, the rescaled consistency index is 0.407, the retention index (RI) is 0.652 and the homoplasy index (HI) is 0.377. Figure 13 shows the 50% majority-rule consensus tree, which indicates the same Desmostylian topology as Matsui & Tsuihiji (2019), except for Paleoparadoxiidae. The Akan specimens are placed as the derived paleoparadoxiids together with P. tabatai, N. repenningi and N. cecilialina. AMP AK960241, N. repenningi and N. cecilialina form a monophyletic clade. AMP AK970253, AMP AK000247 and P. tabatai form a monophyletic clade in which AMP AK970253 is most closely related to P. tabatai. This topology of derived paleoparadoxiids is not supported by the bootstrap value.

Figure 11 Lateral view of the mandible of Paleoparadoxia sp. (AMP AK000247).

(A) Photo. (B) Corresponding line drawing with anatomical interpretations. Hatched areas indicate broken surfaces.

Figure 12 Medial view of the mandible of Paleoparadoxia sp. (AMP AK000247).

(A) Photo. (B) Corresponding line drawing with anatomical interpretations. Hatched areas indicate broken surfaces.

Identification

The Akan specimens exhibit a mosaic of characters shared with Paleoparadoxia and Neoparadoxia. AMP AK960241 exhibits a majority of characters that are diagnostic of Neoparadoxia: an extremely large body size; the paroccipital process is massive, thickened and expand laterally; partially large and deeply concave temporal fossa; a large mandibular fossa. In contrast, this specimen shares the following characters with Paleoparadoxia: the paroccipital process extends slightly medially, and the external occipital protuberance is moderately developed. The 50% majority-rule consensus tree shows that AMP AK960241 and Neoparadoxia form a monophyletic clade. Based on these results, we identify this specimen as Neoparadoxia sp. This is the first record of Neoparadoxia from the western Pacific coast. In addition, the morphological features of AMP AK960241 do not entirely correspond to those of the holotype specimens of N. cecilialina (LACM150150) and N. repenningi (UCMP81302), suggests that it may represent a previously unrecognized species of Neoparadoxia. However, their fragmentary condition requires careful assessment prior to species-level identification.

On the other hand, AMP AK970253 and AMP AK000247 exhibits the majority of characters shared with Paleoparadoxia: the zygomatic arch inclined cranially and lacks dorsoventral expansion; the supraorbital process is slightly developed; the dorsal surface of cranium between supraorbital processes is not depressed; the orbit is located slightly dorsally; the mandibular fossa is smaller compared to N. cecilialina; the coronoid crest of the dentary curved anteriorly; the mandibular symphysis is slightly rotated anteroventrally; the relatively lower position of the mandibular condyle and mandibular foramen; the mandibular body is straight unlike N. repenningi; the dorsal side of the mandibular symphysis lacks foramina, distinguishing it from N. cecilialina. In contrast, these specimens possess the following characters of Neoparadoxia: numerous nutrient foramina with sulci on the dorsal surface of the supraorbital process; the small angle between the anterior and posterior margins of the coronoid process. The 50% majority-rule consensus tree indicates that AMP AK970253, AMP AK000247 and P. tabatai form a monophyletic clade. Therefore, we identify these specimens as Paleoparadoxia sp. However, as their morphological characters do not fully correspond to those of the neotype of P. tabatai (NMS PV-5601), suggesting that they may represent a different species. Given their fragmentary condition, cautious evaluation is necessary before making a species-level identification.

Table 2 Measurements (in mm) of skull of Paleoparadoxia sp. (AMP AK970253).

	AMP AK970253	P. tabatai	N. cecilialina	
Total length (sagittal direction)	185	482	553	
Parietal length	58	54	115	
Frontal length	48a,b	66	145	
Zygomatic width	161a,b	239	329	
Notes.

Modified from Inuzuka (2005) and Barnes (2013).

a Broken, measured as preserved.

b Broken, the measurements are doubled for presentation.

Table 3 Measurements (in mm) of mandible of Paleoparadoxia sp. (AMP AK000247).

	AMP AK000247	P. tabatai	N. repenningi	N. cecilialina	
Total length (sagittal direction)	242a	364	514	418	
Length of cheek tooth alveolar row (p2-m2)	107	118	156	146	
Height of dentary at coronoid process	175	197	296	226	
Transverse width of mandibular condyle	43	44	59	79	
p2 mesiodistal length/buccolingual width	11.4/–	11.2/9.5	–/–	21.3/23.7	
p3 mesiodistal length/buccolingual width	6.2/5.5	16.5/13	–/–	–/–	
p4 mesiodistal length/buccolingual width	9.5/7.1	20/15.5	–/–	21/17	
m1 mesiodistal length/buccolingual width	15.3/11.2	13/12a	–/–	21.8/15	
m2 mesiodistal length/buccolingual width	25.2/12.3	28.5/25	–/–	33.4/25.9	
m3 mesiodistal length/buccolingual width	–/–	34.5/25.5	–/–	–/–	
Notes.

Modified from Inuzuka (2005) and Barnes (2013).

a Broken, measured as preserved.

Figure 13 Time-calibrated fifty percent majority-rule consensus tree showing the phylogenetic relationships of Neoparadoxia sp. (AMP AK960241) and Paleoparadoxia sp. from Akan (AMP AK970253, AMP AK000247).

Fifty percent majority-rule consensus tree resulting from 136 most parsimonious trees, with the consistency index = 0.623 and the retention index = 0.652. Numbers below nodes indicate bootstrap values (10,000 replicates). The values lower than 50% were omitted. The interspecific relationships within clades Desmostylia, Desmostyloidea, Paleoparadoxiidae, and Desmostylidae were omitted and these groups were collapsed to families/superfamilies.

Estimation of desmostylian diversity

The results of the stage-binned analysis are shown in Figs. 14A and 14B. For Paleoparadoxiidae (Fig. 14A), species diversity remained low, with only one or two species (P. tabatai and A. weltoni) present from the Oligocene Chattian to the Miocene Burdigalian (28.1-16 Ma). However, during the Middle Miocene Langhian (16-13.8 Ma), species diversity increased sharply to five species, coinciding with the appearance of the Akan paleoparadoxiids and Neoparadoxia. Subsequently, diversity declined from the Serravallian to the Tortonian (13.8-7.2 Ma), culminating in the complete extinction of Paleoparadoxiidae by the Late Miocene Messinian. In contrast, for Desmostylidae (Fig. 14B), species diversity exhibited minimal fluctuation, resulting in a more stable pattern. From the Oligocene Rupelian to the Chattian (33.9-23 Ma), two basal desmostylids species (A. laticosta and C. sookensis) were present. During the Early Miocene Aquitanian (23-20.4 Ma), A. laticosta and C. sookensis went extinct, and the emergence of more derived desmsotlyids (Ounalashkastylus tomidai and D. hesperus) increased the species count to three, marking a peak in diversity. From the Burdigalian to the Serravallian (20.4-11.6 Ma), despite the extinction of O. tomidai, the appearance of D. japonicus and D. coalingensis maintained the number of species at two. By the Late Miocene Tortonian (11.6-7.2 Ma), only a single species of D. hesperus remained, and like Paleoparadoxiidae, Desmostylidae became completely extinct by the Messinian (7.2-5.3 Ma).

Figure 14 Diversity estimation of Desmostylia.

(A) The stage-binned analysis of Paleoparadoxiidae. (B) The stage-binned analysis of Desmostylidae. (C) The richness curve analysis of Paleoparadoxiidae. (D) The richness curve analysis of Desmostylidae. In the richness curve analysis, maximum (red) and minimum (blue) lineage counts accounting for ghost lineages and stratigraphic uncertainty. Overall analysis based on our chronostratigraphic assessment (Supplemental Information 1). Abbreviations: EOT, the Eocene-Oligocene Transition; OMT, the Oligocene-Miocene Transition; MMCO, the Middle Miocene Climatic Optimum; MMCT, the Middle Miocene Climate Transition.

The results of the richness curve analysis are shown in Figs. 14C and 14D. For Paleoparadoxiidae (Fig. 14C), the maximum and minimum richness curve exhibited similar trends. Both curves gradually increased from the Oligocene Chattian to Aquitanian, with high richness persisting throughout this period (28.1-20.4 Ma). The peak richness was reached during the mid-Burdigalian to Langhian (approximately 18-15 Ma). Since the Langhian, both curves have shown a sharp decline, with possible extinction suggested by the Late Miocene Tortonian (11.6-7.2 Ma). For Desmostylidae (Fig. 14D), the maximum and minimum richness curves also exhibited similar patterns, particularly from the Late Oligocene to the Miocene. The maximum richness curve indicated a gradual increase in richness during the Oligocene, while the minimum curve showed a decrease in richness from the mid-Rupelian to the Early Chattian in the Late Oligocene (approximately 31-27 Ma). Both curves showed an increase in richness after the mid-Chattian, peaking around the Late Chattian and near the Oligocene-Miocene boundary (approximately 25-23 Ma). While both curves ware relatively stable during the Miocene, a stepwise decline was observed at the Middle to Late Burdigalian (approximately 17 Ma) and at the Serravallian-Tortonian boundary (11.6 Ma). Both curves also indicated the extinction of Desmostylidae by the Tortonian (11.6-7.2 Ma).

Discussion

Taxonomic status within Paleoparadoxiidae

Previous studies suggested that two types of Paleoparadoxiidae were present in Akan (Akan vertebrate fossil assemblage research group, 2000; Akan vertebrate fossil assemblage research group, 2002; Inuzuka, 2005). Our results are consistent with previous studies and represent the first record of Paleoparadoxia and Neoparadoxia occurring at the same locality and stratigraphic horizon. The Akan site is characterized by the frequent occurrence of broken fossils, suggesting secondary deposition and the possibility of reworking (Chinzei, 1984; Inuzuka, 2005). Therefore, even fossils discovered from the same stratigraphic level may not have been deposited contemporaneously, and it remains uncertain whether Neoparadoxia and Paleoparadoxia truly coexisted in this region. However, the Middle Miocene was a period of morphological diversification in Paleoparadoxiidae (Inuzuka, 2005; Pyenson & Vermeij, 2016; Matsui, Valenzuela-Toro & Pyenson, 2022), and the morphological analysis of the Akan specimens has the potential to provide valuable insights into the cladogenesis and diversification of this family.

As discussed in the identification section, the Akan specimens exhibit a mosaic of characters shared with both Paleoparadoxia and Neoparadoxia. In AMP AK960241 (Neoparadoxia sp.), two characters resemble those of P. tabatai: (1) the paroccipital process extends slightly medially; (2) the external occipital protuberance is moderately developed (Barnes, 2013). Similarly, in AMP AK970253 and AMP AK000247 (Paleoparadoxia sp.), exhibit two diagnostic characters of Neoparadoxia are present: (1) numerous nutrient foramina with sulci on the dorsal surface of the supraorbital process (Barnes, 2013); (2) the small angle between the anterior and posterior margins of the coronoid process (Inuzuka, 2005). These findings underscore the need for a comprehensive reexamination of the diagnostic characters of both genera. In addition, the Akan specimens do not fully correspond to the morphological characters of the neotype of P. tabatai or the holotypes of Neoparadoxia, and the topology of derived Paleoparadoxiidae is not supported by the bootstrap values. These results indicate that the Akan specimens are difficult to identify at the species level within the existing taxonomic framework of Paleoparadoxiidae, requiring a reevaluation of their phylogenetic hypothesis and diagnostic characters (Kohno, 2024).

Paleodiversity insights of Desmostylia

The stage-binned analysis and the richness curve analysis revealed new insights into the evolutionary history of desmostylian diversification. For Paleoparadoxiidae (Figs. 14A, 14C), during the Langhian of the Middle Miocene (16-13.8 Ma), the stage-binned analysis shows an increase in the number of species, attributed to the appearance of Akan paleoparadoxiids and Neoparadoxia. The richness curve analysis suggests that high richness revel was sustained from the Oligocene Chattian through the Miocene Aquitanian (approximately 28.1-20.4 Ma), and peaked from the Mid-Burdigalian to Langhian (approximately 18-15 Ma). This diversity peak is roughly coinciding with the Middle Miocene Climatic Optimum (MMCO; 16.9-14.7 Ma), a global warming event (e.g., Zachos, Dickens & Zeebe, 2008; Tripati, Roberts & Eagle, 2009). Several marine mammal lineages diversified during the MMCO, including Cetaceans and Pinnipediamorpha (Marx & Fordyce, 2015; Boessenecker & Churchill, 2018; Guo & Kohno, 2023). Previous studies suggested that Paleoparadoxia preferred subtropical to warm-temperate zone (Chinzei, 1984; Itoigawa, 1984; Ogasawara, 2000; Inuzuka, 2005), and the rise in sea level associated with global warming may have expanded the shallow water habitats suitable for Paleoparadoxiidae. Therefore, the MMCO may have promoted the diversification of Paleoparadoxiidae. Following the MMCO, both analyses indicate a significant decline in species diversity during the mid-Langhian to Serravallian (approximately 14.5-11.6 Ma). This decline coincides with the Middle Miocene Climate Transition (MMCT; −14 Ma), a global cooling event associated with the expansion of the East Antarctic Ice Sheet (EAIS) and accelerated cooling (e.g., Flower & Kennett, 1994; Hamon et al., 2013; Frigola, Prange & Schulz, 2018). This climate shift is considered a contributing factor to extinctions across various taxa (Lewis et al., 2008; Guo & Kohno, 2023), and may have played a role in the reduced diversity of Paleoparadoxiidae. After the MMCT, the cooling event of the Late Miocene became more severe (e.g., Herbert et al., 2016; Rousselle et al., 2013; Wen et al., 2023), and both analyses suggest that Paleoparadoxiidae may have gone extinct during the Late Miocene Tortonian to Messinian (11.6-5.3 Ma).

For Desmostylidae (Figs. 14B, 14D), the stage-binned analysis indicates that two basal species, A. laticosta and C. sookensis, appeared during the Oligocene. Near the Oligocene-Miocene boundary, the emergence of O. tomidai and D. hesperus led to a peak in species diversity during the Early Miocene Aquitanian (23-20.4 Ma). The richness curve analysis shows a gradual increase in both maximum and minimum richness curves toward the late-Chattian (approximately 26-23 Ma). These results suggest that the diversity of Desmostylidae may have peaked from the Late Oligocene to the Early Miocene. Around the Oligocene-Miocene boundary (approximately 23 Ma), a brief glacial event known as the Oligocene-Miocene Transition (OMT; e.g., Miller, Wright & Fairbanks, 1991; Zachos, Flower & Paul, 1997; Greenop et al., 2019) occurred, coinciding with ocean current changes, including gradual development of the modern Antarctic Circumpolar Current. The OMT had been associated with faunal turnovers (Marx & Fordyce, 2015; Marx, Fitzgerald & Fordyce, 2019; Deng et al., 2021). Previous studies suggest that Desmostylidae preferred cooler environments (Chinzei, 1984; Itoigawa, 1984; Ogasawara, 2000), with some species discovered in cold polar regions (Ijiri & Kamei, 1961; Beatty, 2006; Chiba et al., 2016). Also, the emergence of kelp ecosystems during the Oligocene has been proposed as a factor that influenced the evolution of marine mammals (Kiel et al., 2024; Tsai, Goedert & Boessenecker, 2024). Desmostylids may likewise have exploited these food resources, contributing to their diversification. Taken together, these findings suggest that the OMT may have driven an increase in diversity of Desmostylidae. From the Early to Middle Miocene (Burdigalian to Serravallian; 20.4-11.6 Ma), species diversity remained relatively stable in both analyses, showing little impact from the MMCO or MMCT. However, as with Paleoparadoxiidae, their species diversity declined during the Serravallian to the Tortonian (13.8-7.2 Ma) and is suggested to have gone completely extinct by the Messinian.

These results indicate that desmostylian diversity may have been closely linked to climate change. This study identified three major shifts in species diversity: (1) Desmostylidae reached peak diversification at the OMT, coinciding with a glacial event; (2) Paleoparadoxiidae experienced its highest diversity during the MMCO, a global warming event; (3) Both families experienced declining diversity and eventual extinction during the Middle to Late Miocene global cooling period. The interaction of desmostylian diversity and these climatic events has not yet been proven to be the direct cause of their rise and fall. Nevertheless, the vicissitudes of desmostylian diversity were obviously influenced by these climatic changes. Interestingly, Desmostylidae and Paleoparadoxiidae exhibited distinct responses to climate change. Desmostylidae diversified during the Oligocene-Miocene boundary event, while Paleoparadoxiidae diversified during the MMCO. This differences likely reflects ecological distinctions, with Paleoparadoxiidae favoring warmer climates and Desmostylidae colder ones (Chinzei, 1984; Itoigawa, 1984; Ogasawara, 2000; Inuzuka, 2005).

However, the reason why Desmostylia exhibited a decline in diversity only during the Middle to Late Miocene cooling event remains uncertain. Previous studies suggested that Desmostylia had successfully adapted to cold environments in other periods (Chinzei, 1984; Matsui & Kawabe, 2015). One possible explanation is that the Middle to Late Miocene cooling was more severe than previous cooling events (Herbert et al., 2016; Rousselle et al., 2013: Wen et al., 2023), their survival more difficult. Another potential factor is ecological competition with Sirenia for food resources (Vélez-Juarbe, Domning & Pyenson, 2012; Pyenson & Vermeij, 2016; Matsui, Valenzuela-Toro & Pyenson, 2022). Berta & Lanzetti (2020) noted that the appearance of the genus Hydrodamalis and other dugongids in the Tortonian (11.6 Ma-) coincides with the decline of desmostylians. While Desmostylia and Sirenia are known to have coexisted in the United States (Parham, Barron & Vélez-Juarbe, 2022), the extent of competitive replacement between these groups in the western Pacific remains poorly understood and requires further investigation.

These paleodiversity estimates from the fossil record may also be influenced by three factors: (1) The volume of fossiliferous rock (Crampton et al., 2003; Smith & McGowan, 2007). Rock volume is strongly correlated with the richness of higher taxa in the Phanerozoic (Hannisdal & Shanan, 2011). Addressing and correcting this bias is essential for obtaining more accurate estimates of paleodiversity (Smith & McGowan, 2007; Uhen & Pyenson, 2007; McGowan & Smith, 2008). However, previous studies have suggested that rock volume containing fossils has a limited impact on marine mammal diversity (Uhen & Pyenson, 2007; Marx, 2009). Since desmostylians are confined to over 200 localities around the North Pacific within a relatively short temporal range, geological biases are likely minimal in these analyses. (2) Taphonomic bias within Desmostylia. Differences in bone microstructure may affect fossil preservation (Ando & Fordyce, 2014). Hayashi et al. (2013) revealed that Paleoparadoxia, Ashoroa, and Behemotops show increased bone mass, while Desmostylus has a spongy inner structure, reflecting secondary aquatic adaptation. Spongy bones may be less likely to be preserved in deposits, which is consistent with the fact that paleoparadoxiids are abundant in Akan while Desmostylus are relatively rare. However, in Utanobori, Hokkaido, Japan, Desmostylus is the only desmostylian taxon found (Inuzuka, 1988; Uno, Kaneko & Takabatake, 2016), and Desmostylus and paleoparadoxiids rarely co-occur in the same formation (Matsui, Valenzuela-Toro & Pyenson, 2022). This suggests that differences in bone microstructure may have less influence on fossil preservation than habitat or ecological factors. While, Hayashi et al. (2013) suggest Paleoparadoxia inhabited shallow while Desmostylus could swim offshore based on bone microanatomical differences, Matsui et al. (2017) argued the opposite based on stratigraphic evidence. These conflicting hypotheses make it difficult to assess the ecological impact on paleodiversity estimates. (3) Taxonomic uncertainty of Desmostylia (Matsui & Pyenson, 2023; Kohno, 2024). Desmostylians have lower species diversity than other marine mammals (Berta & Lanzetti, 2020), and in these analyses, the origination or extinction of a single species had a remarkably large impact on the overall results. As previously mentioned, reassessment of the Desmostylian phylogenetic hypothesis is required, and the results derived from phylogenetic topology analyses should be carefully examined and discussed.

However, these analyses using the existing taxonomic framework provides valuable baseline for understanding Desmostylian diversity history. In the future, a taxonomic reassessment of Desmostylia and more comprehensive phylogenetic analysis will be necessary to better elucidate the drivers of their evolutionary patterns.

Conclusions

We described three paleoparadoxiids specimens from the Middle Miocene Tonokita Formation in Akan, Hokkaido, Japan. Based on the morphological comparisons and phylogenetic analysis, we identified these specimens as Paleoparadoxia sp. and Neoparadoxia sp. This represents the first record of the co-occurrence of two genera of Paleoparadoxiidae, with the potential to provide valuable insights into cladogenesis and morphological diversification within this family. The Akan specimens exhibit mosaic characteristics of Paleoparadoxia and Neoparadoxia. This finding indicates that species-level identification is difficult within the current taxonomic framework of Paleoparadoxiidae, highlighting the need for a reassessment of their phylogenetic hypothesis and diagnostic characters. Additionally, our stage-binned analysis and richness curve analysis shed light on the species-level diversity patterns in Desmostylia. These analyses revealed three significant points in their diversification history: (1) Desmostylidae reached peak diversification at the Oligocene-Miocene boundary, coinciding with a glacial event; (2) Paleoparadoxiidae, achieved peak diversity during the Middle Miocene global warming event; (3) Both families declined their diversity and gone out during the Middle to Late Miocene global cooling event. These finding indicate that desmostylian diversity could have been closely linked to climatic events, with the differing peak diversities of Paleoparadoxiidae and Desmostylidae reflecting their respective preferences for warmer and cooler climates. However, the uncertain taxonomic status of Desmostylia may have influenced diversity estimates. While our analyses provide a valuable baseline for understanding the diversity history of Desmostylia, future efforts should focus on taxonomic reassessment and comprehensive phylogenetic analysis to refine diversity patterns and clarify their evolutionary drivers.

Supplemental Information

Supplemental Information 1 List of valid taxa with chronostratigraphic assessment

Supplemental Information 2 Data matrix

We express our gratitude to Tatsuya Shinmura (AMP) for providing the 3D data of the specimens and granting us access to study them under his care. We are also indebted to the Akan vertebrate fossil investigation team for their dedicated excavation of paleoparadoxiids specimens and locality surveys. Special thanks go to the staff of the AMP for their overall support, as well as to the Kushiro City Board of Education, Hokkaido, Japan, for granting access to the specimens.

Institutional Abbreviations

AMP Ashoro Museum of Paleontology, Hokkaido, Japan

NSM-PV Fossil vertebrate collections at the National Museum of Nature and Science, Tsukuba, Japan

LACM Los Angeles County Museum, California, USA

UCMP University of California Museum of Paleontology, California, USA

Additional Information and Declarations

Competing Interests

Author Contributions

Data Availability

The authors declare there are no competing interests.

Yuma Asai conceived and designed the experiments, performed the experiments, analyzed the data, prepared figures and/or tables, authored or reviewed drafts of the article, and approved the final draft.

Tatsuro Ando conceived and designed the experiments, performed the experiments, analyzed the data, authored or reviewed drafts of the article, and approved the final draft.

Hiroshi Sawamura conceived and designed the experiments, authored or reviewed drafts of the article, and approved the final draft.

Shoji Hayashi conceived and designed the experiments, performed the experiments, authored or reviewed drafts of the article, and approved the final draft.

The following information was supplied regarding data availability:

The three physical paleoparadoxiids specimens are housed at the Ashoro Museum of Paleontology (AMP), Ashoro, Hokkaido, Japan (Neoparadoxia sp. (AMP AK960241): Paleoparadoxia sp. (AMP AK970253, AMP AK000247)).

The data matrix is available in the Supplemental File.

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
