# Peer review of "New evidence for the co-occurrence of two genera of Paleoparadoxiidae (Mammalia, Desmostylia) from the Middle Miocene of Japan: insights into taxonomic status and paleodiversity in Desmostylia"

_PeerJ, doi:10.7717/peerj.19578_

## Round 0.1 · original submission · Minor Revisions

All three reviewers think this work is very useful to our knowledge of Desmostylians, and that this is a high quality paper. The reviewers have included some comments / suggestions to be incorporated that I think will improve the clarity of this paper. Two reviewers note that some spelling / grammar help would be recommended (particularly around scientific terms). Spelling / grammar checking software, including AI tools, might help with this for clarity. Additional comments from the reviewers are attached.

**Language Note:** The review process has identified that the English language must be improved. PeerJ can provide language editing services - please contact us at [email protected] for pricing (be sure to provide your manuscript number and title). Alternatively, you should make your own arrangements to improve the language quality and provide details in your response letter. – PeerJ Staff

·

Basic reporting

This is a significant contribution to our knowledge of desmostylians and is quite professionally executed. The authors are thoroughly acquainted with the relevant literature, the new locality is welcome and worth further prospecting in the future. The cranial specimens are significant contributions to our knowledge, and the descriptions and comparisons of the new fossils are admirably thorough and of high quality, and well labeled. The illustrations overall are very good.

Experimental design

I am satisfied with the authors’ conclusions. I am not familiar with all of their techniques of phylogenetic analysis, but perhaps other reviewers will be better able to comment on those. But I see nothing in the results that concerns me.

Validity of the findings

.

Additional comments

LANGUAGE AND GRAMMAR: This is the one area where serious work is needed. The authors’ English is basically decent, but the text is full of minor typographical errors in punctuation, spelling, and usage. I spotted one representative mistake outside the main text (in Table 1, where the word condyle is missing its final e). Fortunately, the many errors of this sort can be easily identified and fixed by a computer’s spellcheck function, backed up by an editor who is accustomed to how anglophone anatomists describe bones.

On the whole, the authors are to be congratulated on a very solid piece of work, done under the handicap of reading and writing in a very foreign language! I look forward to seeing it in print.

Reviewer 2 ·

Basic reporting

The authors described new paleoparadoxiid specimens from the Middle Miocene of Hokkaido. New materials suggested the co-occurrence of two paleoparadoxiids, and the authors further assessed the large-scale evolution of desmostylians. I recommend the publication of this manuscript, and below are some of my comments/suggestions.

1. Taxonomy and diversity
The authors identified three new specimens belonging to Neoparadoxia sp. (AMP AK960241) and Paleoparadoxia sp. (AMP AK970253 and AMP AK000247). Interestingly, the authors further pointed out that Neoparadoxia sp. (AMP AK960241) has features of Paleoparadoxia, while Paleoparadoxia sp. (AMP AK970253 and AMP AK000247) possess characters of Neoparadoxia. Those observations should, to a large degree, be enough to name new species. The authors seemed conservative in terms of naming new species, but they authors may consider this possibility, as it will also affect the analysis to assess the paleoparadoxiid diversity.

In the Results section, the authors gave a separate Identification for the new paleoparadoxiid specimens. For clarity, it may be better to include this in the Systematic Paleontology section. For example, the authors can add a Remarks section to emphasize those interesting combinations of morphological characters if they are not naming new species.

2. Ecological niche and food resources
The authors attempted to correlate the desmostylian diversity to the climatic events. The climate is indeed critical, but the authors may like to discuss a bit more about other ecological aspects, such as niche partitioning and food resources. Desmostylians were herbivores and primarily fed on seagrass. A new publication discovered the fossil kelp (seagrass) from the Early Oligocene, and the origin of the kelp ecosystem likely shaped the desmostylian evolution (Kiel et al. 2024 PNAS). The kelp ecosystem fosters superabundant food resources, and this has also been linked to the early evolution of other marine mammals (Mysticeti: Tsai et al. 2024 Current Biology).

Similarly, the authors briefly mentioned the possible competition between desmostylians and sirenians. In this manuscript, it seems highly relevant to discuss the possibility of how two genera of paleoparadoxiids coexisted. For example, niche partitioning is one straightforward viewpoint. The authors briefly mentioned the size differences between the two genera of paleoparadoxiids from this locality, but it would be helpful to estimate the body size of each paleoparadoxiid. Given the size difference and a relevant example of fossil marine mammals from Hokkaido, there was a previous publication that demonstrated the size disparity of toothed mysticetes and discussed the potential niche partitioning (Tsai and Ando 2016 Journal of Mammalian Evolution).

3. Other minor points
It is inevitable that there are some typos, but for technical terms, we always need to be more careful. For example, the authors used “paraoccipital process”, but it should be “paroccipital process”.

Speaking of the climate fluctuations and extinction of marine species during the Miocene, there was a Science paper (Sibert and Rubin 2021 Science), which suggested a large-scale extinction in pelagic sharks during the Early Miocene. This paper was criticized quite a bit, but the authors may like to read it and think of how it may relate to the desmostylian evolution.

Regards,

Experimental design

-

Validity of the findings

-

·

Basic reporting

-

Experimental design

-

Validity of the findings

-

Additional comments

The article is of great interest and presents a very complete study on the subject, with significant findings on the current issues that need clarification and new data surrounding the evolution of Desmostylia.

---

## Round 0.2 · accepted · Accept

Thanks for the revision; I have assessed the revision and think you have addressed all of the reviewers' comments. Congratulations on publication!